# Culturing the Chicken Intestinal Microbiota and Potential Application as Probiotics Development

**DOI:** 10.3390/ijms24033045

**Published:** 2023-02-03

**Authors:** Ke Ma, Wei Chen, Xiao-Qi Lin, Zhen-Zhen Liu, Tao Wang, Jia-Bao Zhang, Jian-Gang Zhang, Cheng-Kai Zhou, Yu Gao, Chong-Tao Du, Yong-Jun Yang

**Affiliations:** Key Laboratory of Zoonosis Research, Ministry of Education, College of Veterinary Medicine, Jilin University, Changchun 130012, China

**Keywords:** chicken, intestinal microbiota, culturing, probiotic development, anti-*S*. Pullorum infection

## Abstract

Pure cultures of chicken intestinal microbial species may still be crucial and imperative to expound on the function of gut microbiota, and also contribute to the development of potential probiotics and novel bioactive metabolites from gut microbiota. In this study, we isolated and identified 507 chicken intestinal bacterial isolates, including 89 previously uncultured isolates. Among these, a total of 63 *Lactobacillus* strains, belonging to *L. vaginalis*, *L. crispatus*, *L. gallinarum*, *L. reuteri*, *L. salivarius*, and *L. saerimneri*, exhibited antibacterial activity against *S. Pullorum*. Acid tolerance tests showed *Limosilactobacillus reuteri* strain YPG14 (*L. reuteri* strain YPG14) has a particularly strong tolerance to acid. We further characterized other probiotic properties of *L. reuteri* strain YPG14. In simulated intestinal fluid, the growth of *L. reuteri* strain YPG14 remained stable after incubation for 4 h. The auto-aggregation test showed the auto-aggregation percentage of *L. reuteri* strain YPG14 was recorded as 15.0  ±  0.38%, 48.3  ±  2.51%, and 75.1  ±  4.44% at 3, 12, and 24 h, respectively. In addition, the mucin binding assay showed *L. reuteri* strain YPG14 exhibited 12.07 ±  0.02% adhesion to mucin. Antibiotic sensitivity testing showed that *L. reuteri* strain YPG14 was sensitive to the majority of the tested antibiotics. The anti-*Salmonella* Pullorum (*S*. Pullorum) infection effect in vivo revealed that the consumption of *L. reuteri* strain YPG14 could significantly improve body weight loss and survival rate of chicks infected by *S*. Pullorum; reduce the loads of *S*. Pullorum in the jejunum, liver, spleen, and feces; and alleviate the jejunum villi morphological structure damage, crypt loss, and inflammatory cell infiltration caused by *S*. Pullorum. Overall, this study may help us to understand the diversity of chicken intestinal microflora and provide some insights for potential probiotic development from gut microbiota and may find application in the poultry industry.

## 1. Introduction

The extensive research on the microbiome of the chicken gastrointestinal tract (GIT) has repeatedly vindicated that the gut microbiota plays an important role in the chicken’s health, growth, and development [1]. The gut microbiota has gradually become a promising source of probiotics and novel bioactive metabolites [2,3,4], except for nutrient intake, digestion, immune modulation, and the exclusion of pathogens. However, unfortunately, a large proportion of these gut microbiota are not yet cultured, which limits our understanding of the interactions between individual members of the gut microbiota and hosts [5]. Promisingly, some recently developed approaches that isolate uncultivated microorganisms have proven successful [6].

*Lactobacillus* strains are important members of the human and animal gut microbiota and are also among the most widely used probiotics because of their great advantages [7]. Because of the ban on antibiotics in livestock production, a *Lactobacillus*-dominated probiotic was proposed as a possible alternative [8]. A wide variety of *Lactobacillus* strains have been used as potential probiotics in the growth performance and immune response of poultry [9]. However, for cross-species application of probiotics, it cannot be ignored that we need to recognize the target host-microbial species’ nature and, therefore, it is advisable to source potential target host probiotic strains from the target host intestine [10].

Here, we work on culturing the chicken intestinal microbiota by referencing the culturomics method and screening the *Lactobacillus* strains with potential probiotic properties. We further characterize the probiotic properties and colonization resistance of a promising probiotic candidate strain *L. reuteri* strain YPG14. These results allowed us to understand the diversity of chicken intestinal microbiota, and their potential application as probiotic development in the poultry industry.

## 2. Results

### 2.1. Microbial Diversity of Chicken Intestine

A total of 507 pure cultures were isolated from the intestine of chicken (Appendix A). Based on the 16S rRNA gene sequence, it represented 41 different species from 17 genera belonging to the phyla *Firmicutes*, *Bacteroidetes*, and *Proteobacteria* (Figure 1A). It is dominated by *Firmicutes* (74.8% of strains), and *Lactobacillus* (62.3% of strains). In addition, a total of 89 strains were characterized as previous uncultured bacteria (Figure 1B), and based on the recommended 16S rRNA sequence 98.65% similarity thresholds for identifying bacterial species [11], it could be assigned potential for four novel species, among them, *Paenibacillus jilinensis* sp. nov. YPG26 has been validated in our previous studies [12], but we are also working to validate three additional novel species by whole genome sequencing-based phylogenomic analysis. To further investigate the effect of mediums on bacterial isolates, we analyzed and summarized the number of isolates from each medium; the results showed that the species of bacterial isolates increased significantly in the M2 and M3 medium supplemented with FBS or blood, respectively (Table 1).

### 2.2. Selection of Lactobacillus Strains with Antimicrobial Activity against Salmonella Pullorum

A total of 63 *Lactobacillus* strains were selected to determine the antimicrobial effects of the CFS against *Salmonella* Pullorum ATCC19945. All selected *Lactobacillus* strains, including *L. vaginalis*, *L. crispatus*, *L. gallinarum*, *L. reuteri*, *L. salivarius*, and *L. saerimneri*, exhibited antibacterial activity against *S*. Pullorum ATCC19945 (Figure 2). Among these strains, some *Lactobacillus* strains were selected due to their good antibacterial activity against *S*. Pullorum.

### 2.3. Acid Tolerance Properties of the Selected Lactobacillus Strains

Following the above antibacterial analysis, some *Lactobacillus* strains were selected due to their good antibacterial activity against *S*. Pullorum, including the strains Y48, Y63, Y40, Y49, Y67, Y185, Y187, Y313, YPG14, YPG16, WCB43, YPG21, WCB24, and WCB2, and further evaluated for acid tolerance capacity. The results showed that all strains could survive at pH 2.0 after 60 and 90 min of incubation, but the strains YPG14 and YPG16 have a particularly strong tolerance to acid (Figure 3). Combining good antibacterial activity and acid tolerance capacity, the strain YPG14 was therefore selected for subsequent evaluation of its probiotic properties.

### 2.4. Identification of the Strain YPG14 and Growth Time Curve

The neighbor-joining and maximum likelihood phylogenetic tree based on the 16S rRNA gene sequences of the strain YPG14 (1445 bp) showed the strain YPG14 was closest related to the species *Limosilactobacillus reuteri* subsp. *reuteri* strain DSM 20016 (GenBank accession no. NR_075036.1) with 99.79% similarity (Figure 4); hence, the strain YPG14 belonged to *Limosilactobacillus reuteri*. The accession number for the 16S rRNA gene sequence of the strain YPG14 deposited in the GenBank database is OP953762.

Further, we also characterized the growth time curve and the pH value change. The growth of the strain YPG14 in liquid medium assayed by OD_600_ showed that the logarithmic growth phases were 4–24 h (Figure 5A); meanwhile, the pH values of the medium for the strain YPG14 dropped gradually with the increase in culture time and remained relatively stable after 24 h (Figure 5B).

### 2.5. Simulated Intestinal Fluid Tolerance of the Strain YPG14

In simulated intestinal fluid, the growth of the strain YPG14 remained stable after incubation for 4 h (Figure 6A). It showed good tolerance to simulated intestinal fluid.

### 2.6. Auto-Aggregation Test of the Strain YPG14

The auto-aggregation percentage of the strain YPG14 was recorded as 15.0  ±  0.38% at 3 h incubation. It increased to 48.3  ±  2.51% and 75.1  ±  4.44% at 12 h and 24 h, respectively (Figure 6B). The result demonstrated that the strain YPG14 possessed excellent auto-aggregating properties.

### 2.7. Mucin Binding Assay of the Strain YPG14

A mucin binding assay revealed that the strain YPG14 exhibited 12.07  ±  0.02% adhesion to mucin, and the CFU/mL of the strain YPG14 adhesion to mucin was depicted in Figure 6C.
Figure 6The characterization of partial probiotic properties of the strain YPG14. (**A**) The tolerance of the strain YPG14 to simulated intestinal juice; (**B**) The auto-aggregation percentage of the strain YPG14; (**C**) The adhesion of the strain YPG14 to mucin.
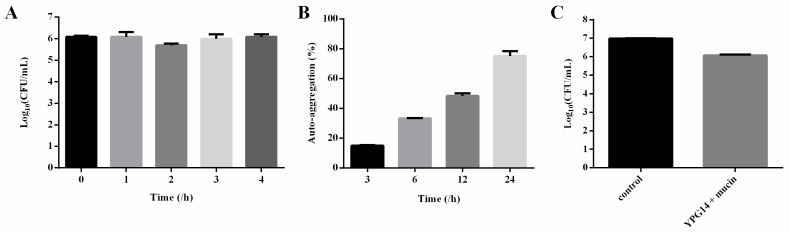



### 2.8. Antibiotic Sensitivity Testing

Antibiotic sensitivity testing by using the disc diffusion method showed that the strain YPG14 was sensitive to the majority of the tested antibiotics except for gentamicin and vancomycin (Table 2).

### 2.9. Survival Rate and Growth Status of Chicks and Enumeration of Salmonella Pullorum

As shown in Figure 7A, a total of 80 newly hatched broiler chickens were randomly divided into four groups. Among them, the YPG14 treatment group (YPG14) and the strain YPG14 and *S*. Pullorum treatment group (*S*. Pullorum +YPG14) were administrated with 1.0 × 10^8^ CFU/bird of the strain YPG14 for 7 consecutive days, and then the *S*. Pullorum treatment group (*S*. Pullorum) and the *S*. Pullorum +YPG14 treatment group were challenged with 1.0 × 10^9^ CFU/bird *S*. Pullorum on the third day. The results showed that the strain YPG14 could significantly inhibit the body weight loss caused by *S*. Pullorum, especially in 5- and 6-day-old chicks (Figure 7B). Moreover, the strain YPG14 treatment significantly improved chicken survival when compared to the *S*. Pullorum-challenged group, especially on the eighth day, as the mortality of chicks in the *S*. Pullorum group was over 80% compared with only 40% in the *S*. Pullorum +YPG14 group (*p* < 0.001), while no deaths were observed in the control and YPG14 groups (Figure 7C).

Furthermore, the loads of *S*. Pullorum in the jejunum, liver, spleen, and feces were determined. In comparison with the *S*. Pullorum group, the number of *S*. Pullorum colonized in the tissues and feces was significantly lower in the *S*. Pullorum +YPG14 group (*p* < 0.05) (Figure 7D).

### 2.10. Histopathology Analysis

Histopathology analysis based on H&E-stained pathological sections revealed that *S*. Pullorum caused severe damage to the villi morphological structure of the jejunum, massive crypt loss, and there was a large number of inflammatory cell infiltration, whereas the strain YPG14 treatment significantly reduced the jejunum injury compared to the *S*. Pullorum group (Figure 8A). The height of the jejunum villi (Figure 8B) and the histopathology score (Figure 8C) in the *S*. Pullorum + YPG14 group also were also significantly lower than those in the *S*. Pullorum group. These results demonstrated a protective role of the strain YPG14 in the *S*. Pullorum infection.

## 3. Discussion

As the most main economic animal in the poultry industry, chicken has become an important source produce of valuable protein for humans. A large microbial community inhabits the intestine of chicken, including broilers [13], layers [14], and feral chickens [15]. The intestinal microflora together with the host maintains its microecological balance and plays a crucial role in chicken digestion, nutrient absorption, pathogen resistance, immune regulation, and development [16]. Unfortunately, most of them are still uncultivable due to certain limiting factors, but some recently developed approaches which isolate uncultivated microorganisms, such as the culturomics method [5], have proven successful. In this study, we isolated and identified a total of 507 pure cultures from the chicken intestine based on 11 isolation media by referencing the culturomics method. These isolates represented 41 different species from 17 genera, and 89 strains were characterized as previous uncultured bacteria. Based on the recommended 16S rRNA sequence 98.65% similarity thresholds for identifying novel bacterial species [11], these uncultured bacteria could be assigned four potential novel species, including *Paenibacillus jilinensis* sp. nov., validated in our previous studies [12], which fully demonstrate the diversity of chicken intestinal microbiota. In addition, these isolates were dominated by *Lactobacillus* (62.3% of strains). This is also consistent with previous research. In general, bacterial communities of the chicken intestine vary considerably by different intestinal segments, but are dominated by the genus *Lactobacillus* in all segments except for the cecum [17].

*Lactobacillus* has been widely reported as probiotics for chicken growth performance and health, and because of the ban on antibiotics in livestock production, *Lactobacillus*-dominated probiotic is also proposed as a possible alternative [8]. The antimicrobial activity has been proposed to be important for the ability of *Lactobacillus* strains to exert probiotic properties on their hosts. In antimicrobial screening, we found that the cell-free supernatants of 63 *Lactobacillus* isolates, including *Lactobacillus vaginalis*, *Lactobacillus crispatus*, *Lactobacillus gallinarum*, *Lactobacillus reuteri*, *Lactobacillus salivarius*, and *Lactobacillus saerimneri*, exhibited antibacterial activity against *S. Pullorum*. The antimicrobial properties emphasized their potential for use as probiotics.

Resistance to the harsh conditions of the GI tract is an important prerequisite for efficient colonization by a probiotic strain [18]. The acid tolerance tests confirmed *L. reuteri*, including the strains YPG14 and YPG16, possessed a high survival rate under an acidic environment (pH 2.0). In addition, in simulated intestinal fluid, the growth of the strain YPG14 remained stable after incubation for 4 h. These results are similar to other *L. reuteri* tolerance reported by others [19]. Therefore, combining good antibacterial activity and tolerance capacity, *L. reuteri* strain YPG14 was selected for subsequent evaluation of probiotic properties.

An auto-aggregation phenotype and mucin binding ability contribute to the persistence of the organism in the GI tract and adhesion to epithelial cells, which further promote the intestinal colonization of the organism and perform probiotic functions [20]. The auto-aggregation percentage of *L. reuteri* strain YPG14 was recorded as 75.1  ±  4.44% at 24 h, and the result indicated that *L. reuteri* strain YPG14 possessed excellent auto-aggregating properties. In addition, the mucin binding assay showed *L. reuteri* strain YPG14 exhibited 12.07 ±  0.02% adhesion to mucin, and these results are also similar to other *Lactobacillus* tolerance reported by others [21]. The high cell surface hydrophobicity and adhesion ability of *L. reuteri* strain YPG14 would be helpful for its intestinal colonization.

One of the safety considerations in probiotic studies is that a potential probiotic strain does not contain transferable resistance genes. Antibiotic sensitivity testing by using the disc diffusion method showed that *L. reuteri* strain YPG14 was sensitive to the majority of the tested antibiotics except for gentamicin and vancomycin. The result was also similar to other research [19]. This indicated that *L. reuteri* strain YPG14 has good safety and does not contain transferable resistances.

As one of the most pathogenic bacteria in chicken, *Salmonella* Pullorum is widely spread and causes great economic losses to the poultry industry [22]. The traditional antibiotics used to prevent *Salmonella* infection could contribute to antibiotic resistance, thus, there is a need to look for alternative approaches. Increasing evidence has indicated that probiotics, particularly Lactobacilli, are a promising alternative to traditional antibiotic in the treatment of anti- *S*. Pullorum infection. Multiple studies also indicated *L. reuteri* can inhibit *S*. Pullorum infection [23,24]. We also found that, as a potential probiotic, the consumption of *L. reuteri* strain YPG14 could significantly improve the health status, body weight loss, and survival rate of chicks infected by *S*. Pullorum. In addition, the loads of *S*. Pullorum in the jejunum, liver, spleen, and feces were significantly reduced after treatment of *L. reuteri* strain YPG14. Histopathology analysis also revealed that *S*. Pullorum caused severe damage to the villi morphological structure of the jejunum, massive crypt loss, and there was a large number of inflammatory cell infiltration, whereas *L. reuteri* strain YPG14 treatment significantly reduced the jejunum injury. These results demonstrated good protective effects of *L. reuteri* strain YPG14 in the *S*. Pullorum infection, however, the specific protective mechanism needs further exploration in the future.

## 4. Materials and Methods

### 4.1. Sample Collection and Processing

The chickens were euthanized by cervical dislocation and the fresh feces, intestinal content, and mucosal samples were collected. The animal handling procedures were conducted according to experimental practices and standards approved by the Animal Welfare and Research Ethics Committee at Jilin University (No. KT202003219). Subsequently, the microorganism samples were transferred to the anaerobic workstation and serially diluted in sterile phosphate-buffered saline (PBS) solution, and plated on 11 different agar plates (see composition below), which were then incubated under anaerobic conditions (85% N_2_, 10% H_2_, 5% CO_2_) for 3 days at 37 °C. The strain was purified by subculturing, and stored at −80 °C in media with 25% glycerol.

### 4.2. Culture Media

A total of 11 different culture media were designed for this research as previously described with modifications [5,25]. All media formulations were prepared as follows, and quantities are per liter of medium.

M1: BHI (Qingdao Hope Biotechnology Co., Ltd., Qingdao, China): 18.5 g, yeast extract (Oxoid, Hampshire, UK): 5 g, TSB (Qingdao Hope Biotechnology Co., Ltd., Qingdao, China): 15 g, K_2_HPO_4_: 2.5 g, glucose: 0.5 g, palladium chloride (Coolaber, Beijing, China): 0.33 g, mucin (Sigma-Aldrich, St Louis, MO, USA): 4 g. After autoclaving, adding hemin (Coolaber, Beijing, China): 10 µg, Trace Metal Mix A5 (Vector Elite, Burlingame, CA, USA): 1 mL, menadione (Coolaber, Beijing, China): 5 µg, biotin (Coolaber, Beijing, China): 10 µg, vitamin B12 (Coolaber, Beijing, China): 10 µg, vitamin B6 (Coolaber, Beijing, China): 100 µg, folic acid (Coolaber, Beijing, China): 50 µg, L-cysteine hydrochloride monohydrate (Source Leaf Creature, Shanghai, China): 0.6 g. M2: M1 + 5% fetal bovine serum (Gibco, Waltham, MA, USA). M3: M1 + 5% sheep blood (Solaibao, Beijing, China). M4: BHI: 38.5 g, yeast extract: 10 g, L-cysteine hydrochloride monohydrate: 1 g, hemin: 15 mg. M5: Reinforced Clostridium Medium (Qingdao Hope Biotechnology Co., Ltd., Qingdao, China). M6: glycerol: 10 mL, L-asparagine (Coolaber, Beijing, China): 1 g, K_2_HPO_4_: 1 g, MgSO_4_.7H_2_O: 0.5 g, CaCO_3_: 0.3 g, menadione: 5 µg, biotin: 10 µg, vitamin B12: 10 µg, vitamin B6: 100 µg, folic acid: 50 µg, Trace Metal Mix A5: 1 mL. M7: beef extract powder: 2.4 g, tryptone: 10 g, L-cysteine hydrochloride monohydrate: 0.6 g, glucose: 2.5 g, yeast extract: 5 g, NaCl:5 g. M8: MRS (Qingdao Hope Biotechnology Co., Ltd., Qingdao, China). M9: MRS: 52.24 g, L-cysteine hydrochloride monohydrate: 0.5 g, NaCl: 5 g, hemin, 10 µg. M10: WCA (Wilkins-Chalgren Anaerobe, Oxoid, Hampshire, UK): 33 g; glucose: 4 g; hemin, 10 µg; L-cysteine hydrochloride monohydrate, 0.4 g. M11: Tryptone (Oxoid, Hampshire, UK): 20 g, Glucose: 5 g, Yeast extract: 10 g, NaCl: 0.08 g, CaCl_2_: 0.008 g, MgSO_4_: 0.008 g, NaHCO_3_: 0.4 g, soluble starch (Solaibao, Beijing, China): 5 g, pectin (Coolaber, Beijing, China): 0.5 g, L-cysteine hydrochloride monohydrate: 0.5 g. All other reagents were purchased from Beijing Chemical Factory (Beijing, China).

### 4.3. 16S rRNA Gene Sequencing and Phylogenetic Analysis

The 16S rRNA gene of the isolates was amplified and sequenced as previously described [26]. In brief, the DNA was extracted from pure bacterial cells, and the 16S rRNA gene was amplified using universal bacterial primers 27F and 1492R. The PCR product was sequenced, and the 16S rRNA gene sequence was compared with sequences available in GenBank by the nucleotide BLAST to determine an approximate phylogenetic affiliation. The phylogenetic tree was set up using the neighbor-joining and maximum likelihood method in the MEGA 11.0 software [27], and the topologies were evaluated using the bootstrap resampling method with 1000 replications.

### 4.4. Selection of Lactobacillus Strains with Antimicrobial Activity against Salmonella Pullorum

The antimicrobial activity of CFS (cell-free supernatants) was evaluated initially by the agar well diffusion method as previously described, with appropriate modifications [28]. Briefly, to prepare CFS, all selected *Lactobacillus* strains were cultured in MRS at 37 °C for 48 h under anaerobic conditions. After incubation, the bacterial suspension was centrifuged at 8000× *g* for 10 min, and supernatants were collected and filter-sterilized with a 0.22 μm filter (Millipore, Billerica, MA, USA). Overnight cultures of *Salmonella Pullorum* ATCC19945 were used as the indicator strain. After incubation at 37 °C for 20 h, the diameters of inhibition zones formed by tested CFS were measured with a vernier caliper. A promising strain with good antibacterial activity was selected for the following probiotic potential studies. Two biological replicates were set. Additionally, the antimicrobial activity of CFS of *Lactobacillus saerimneri* was evaluated using other pathogens (*Staphylococcus aureus* strain USA300 TCH1516 (ATCC BAA-1717) and *Pseudomonas aeruginosa* ATCC 27853).

### 4.5. Acid Tolerance Properties of the Selected Lactobacillus Strains

The acid tolerance experiments were carried out according to the previous method with a few improvements [29]. Briefly, to determine the survival rate of the selected *Lactobacillus* strains under acid conditions, overnight cultures of the *Lactobacillus* strains were inoculated into MRS broth with pH 2 and incubated at 37 °C for 30 and 60 min, and the normal MRS broth was used as a control. After incubation, 100 μL aliquots were taken for tenfold serial dilutions, then 100 µL of the dilutions were plated on the MRS agar plates and incubated overnight at 37 °C, colonies were counted and CFU per mL was calculated.

### 4.6. Determination of Growth Performance of the Strains YPG14

The strain YPG14 was cultured overnight in MRS broth, and then the bacterial suspensions were inoculated at a ratio of 1/100 into MRS broth (30 mL). The OD_600_ and pH were measured every 2 h for a total of 36 h (*n* = 3).

### 4.7. Simulated Intestinal Fluid Tolerance of the Strain YPG14

Simulated intestinal fluid (SIF) was prepared according to Chinese Pharmacopeia (2010) and the method of Yang JM et al. [30]. Concisely, overnight cultures of the strain YPG14 were inoculated into either SIF and incubated at 37 °C under anaerobic conditions. The survivability was determined by counting on MRS agar plates at 0, 1, 2, 3, and 4 h after SIF treatment, and expressed in CFU per mL.

### 4.8. Auto-Aggregation Test of the Strain YPG14

Auto-aggregation of the strain YPG14 was determined as previously described with appropriate modifications [21]. Briefly, overnight cultures of the strain YPG14 were harvested by centrifugation at 8000× *g* for 10 min at 4 °C. The bacterial cells were washed twice with PBS solution and diluted to 1.0 OD (600 nm). The 2 mL suspension was transferred to a 12-well bacterial plate and incubated at 37 °C for 3, 6, 12, and 24 h. After incubation, the upper suspension was removed carefully and the absorbance values at 600 nm were measured using a microplate reader (Eppendorf, Hamburg, Germany). The auto-aggregation percentage was expressed as (OD of 0 h bacterial suspension − OD of upper suspension/OD of 0 h bacterial suspension) × 100.

### 4.9. Mucin Binding Assay of the Strain YPG14

The ability of the strain YPG14 to bind to mucin was evaluated as previously described with appropriate modifications [31]. The 100 μL of mucin at a concentration of 10 mg/mL were added to a 96-well plate and incubated overnight at 4 °C. The plate was washed twice with PBS and saturated with a 2% (*w*/*v*) bovine serum albumin (BSA) solution (Sigma-Aldrich, St Louis, MO, USA) for 4 h at 4 °C. The plate was washed twice with PBS, and then 100 μL YPG14 bacterial suspension was added to each well. The plate was incubated at 37 °C for 1 h. After incubation, the wells were washed 4 times with PBS to remove non-adherent bacteria. The 200 μL of 0.5 % Triton X-100 (Sigma-Aldrich, St Louis, MO, USA) was added to each well and incubated for 2 h at room temperature, then the wells were scraped with a sterile tip and the suspension was taken for tenfold serial dilutions. Then, 100 µL of the dilutions was plated on the MRS agar plates and incubated overnight at 37 °C. Colonies were counted and the CFU per mL was calculated.

### 4.10. Antibiotic Sensitivity Testing

The antibiotic sensitivity of the strain YPG14 was tested by modifying the disc diffusion test as previously described [32]. Briefly, commercial antibiotic discs (Hangzhou Microbial Reagent Co., Ltd., Hangzhou, China) were placed on MH agar (Qingdao Hope Biotechnology Co., Ltd., Qingdao, China) plates inoculated with the strain YPG14(10^8^ CFU). The plates were incubated at 37 °C for 24 h. Inhibition zone diameters were measured and referred to the Clinical and Laboratory Standards Institute interpretative zone diameters for disc diffusion. The results were described in terms of resistance (R), moderate resistance (MR), or sensitivity (S).

### 4.11. Chicks’ Management and Experimental Design

The animal experimental procedures were conducted according to experimental practices and standards approved by the Animal Welfare and Research Ethics Committee at Jilin University (No. KT202003219), and the experimental process was also referred to as previously described with appropriate modifications [33,34]. Briefly, a total of 80 1-day-old broiler chickens were randomly divided into 4 groups: the control group (Control), the strain YPG14 treatment group (YPG14), the *Salmonella* Pullorum ATCC19945 treatment group (*S.* Pullorum), and the strain YPG14 and *S.* Pullorum treatment group (*S.* Pullorum + YPG14). The details of the chick experimental design are illustrated in Figure 7A. In each group, 15 chicks were used to count the body weight and survival rate. On the eighth day, the other 5 chicks were euthanized, and the jejunum segments were sampled.

### 4.12. Enumeration of Salmonella Pullorum

The tissue samples (jejunum, liver, spleen, and feces) were aseptically collected from the euthanized animals at 5 days post-infection (dpi). Samples were homogenized mechanically in cold PBS, then serially tenfold diluted in sterile PBS, and a 50 µL aliquot of the dilutions were plated on the SS agar (Qingdao Hope Biotechnology Co., Ltd., Qingdao, China) plate and incubated overnight at 37 °C. Colonies were counted, and CFU per g was calculated.

### 4.13. Histopathology Analysis and Scoring

The jejunum issue was fixed in a 4% paraformaldehyde buffer and then embedded in paraffin. Subsequently, tissue was cut into slices and stained with hematoxylin and eosin (H&E). Histopathological changes and scores were detected and quantified under light microscopy, and villi height was also measured according to a previous study [35].

### 4.14. Statistical Analyses

Data were analyzed using GraphPad Prism 7.0 and expressed as the means ± SEM. The significance of differences (** p* < 0.05, *** p* < 0.01, **** p* < 0.001) was calculated by one-way ANOVA or a Student’s *t*-test.

## 5. Conclusions

In conclusion, we isolated and identified 507 chicken intestinal bacterial isolates, including some previous uncultured bacteria. It provides insight into the wide diversity of the intestine microbiota of poultry. Additionally, combining good antibacterial activity and tolerance capacity, *L. reuteri* strain YPG14 was selected for subsequent evaluation of probiotic properties. Auto-aggregation and mucin binding further confirmed that *L. reuteri* strain YPG14 displayed good probiotic properties. Antibiotic sensitivity testing showed that *L. reuteri* strain YPG14 was sensitive to the majority of the tested antibiotics. The anti-*S.* Pullorum infection activity in vivo revealed that the consumption of *L. reuteri* strain YPG14 showed good protective effects. Therefore, these features, combined with the GRAS status of Lactobacilli, allow the application of *L. reuteri* strain YPG14 as a potential candidate probiotic. Nevertheless, further specific protective mechanism characterization of *L. reuteri* strain YPG14 in vivo will contribute to understanding the effect on host health and promote its wider application.

## Figures and Tables

**Figure 1 ijms-24-03045-f001:**
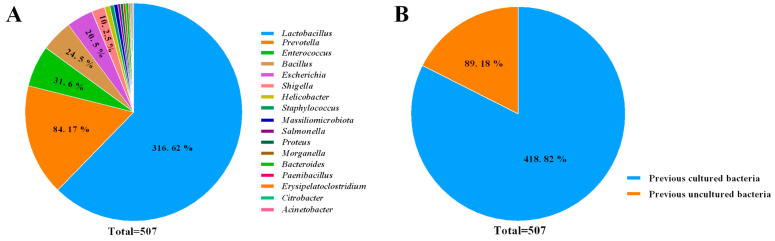
Diversity of the chicken intestinal microbiota strain collection. (**A**) Taxonomic distribution of genus level of total pure cultured bacteria isolated from chicken intestine; (**B**) The percentage of previously cultured and uncultured chicken intestinal bacteria.

**Figure 2 ijms-24-03045-f002:**
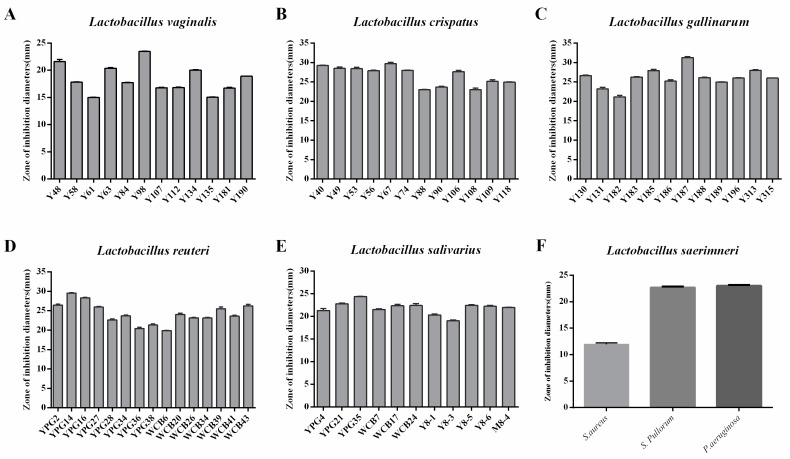
The antimicrobial activity of *Lactobacillus* isolates against *Salmonella* Pullorum. (**A**–**F**) represent the antimicrobial activity of *L. vaginalis*, *L. crispatus*, *L. gallinarum*, *L. reuteri*, *L. salivarius*, and *L. saerimneri*, respectively.

**Figure 3 ijms-24-03045-f003:**
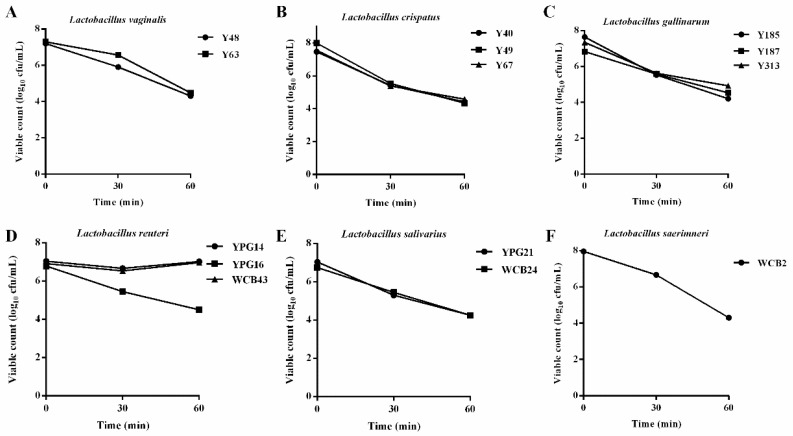
The acid tolerance properties of the selected *Lactobacillus* strains. (**A**–**F**) represent the acid tolerance properties of *L. vaginalis*, *L. crispatus*, *L. gallinarum*, *L. reuteri*, *L. salivarius*, and *L. saerimneri*, respectively.

**Figure 4 ijms-24-03045-f004:**
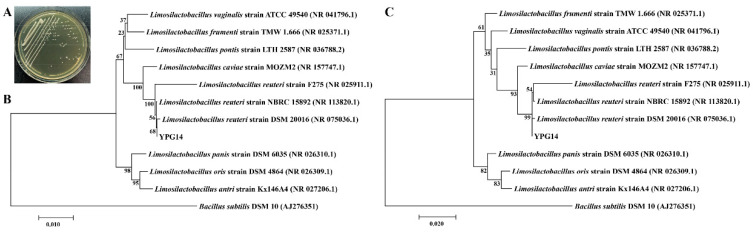
Phylogenetic analysis of the strain YPG14. (**A**) Colony morphology of the strain YPG14; (**B**) Neighbor-joining and (**C**) maximum likelihood phylogenetic tree based on the 16S rRNA gene sequence of the strain YPG14 (1445 bp) showed the taxonomic position of the strain YPG14 and closely related taxa. Bootstrap values (percentages of 1000 replications) are shown at branch points. *Bacillus subtilis* DSM 10T (GenBank accession no. AJ276351) was used as outgroup. The bar, 0.01 and 0.02 nucleotide substitutions per site.

**Figure 5 ijms-24-03045-f005:**
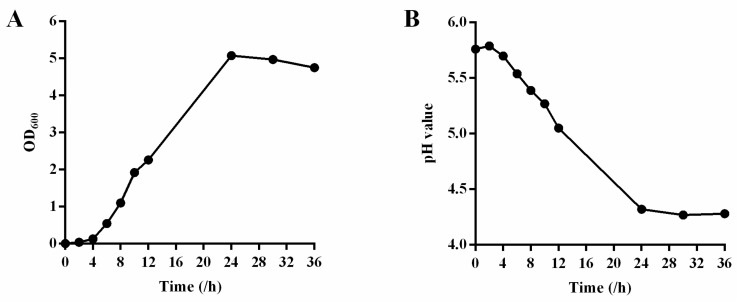
The growth determination of the strain YPG14. (**A**) The growth time curve of the strain YPG14; (**B**) The pH value change of media during growth of the strain YPG14.

**Figure 7 ijms-24-03045-f007:**
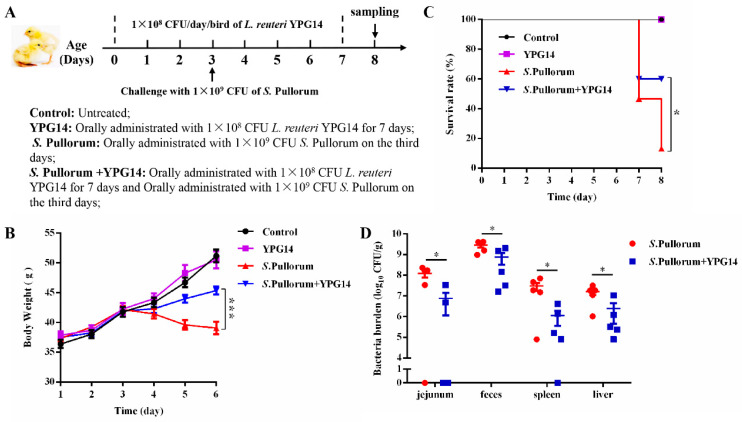
Study design and protective effect of the strain YPG14 against *Salmonella* Pullorum infections. (**A**) The experimental design and treatment procedure; (**B**) The determination of body weight change (*n* = 15/group); (**C**) The determination of survival rate (*n* = 15/group); (**D**) The determination of *S*. Pullorum bacterial burden in tissues and feces infected with *S*. Pullorum on 5-day post-infection (dpi). All data are shown as mean ± SEM. Student’s *t*-test was performed. Statistical significance is indicated by * *p* < 0.05 and *** *p* < 0.001.

**Figure 8 ijms-24-03045-f008:**
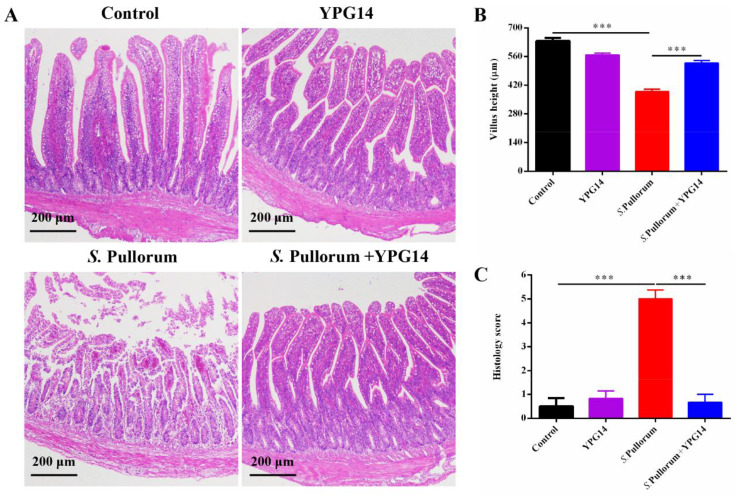
Histopathological changes of jejunum tissues. (**A**) Histopathological changes in jejunum tissues were examined by H&E staining (scale bar = 200 μm); (**B**) The determination of villus length of jejunum tissues (10 villi per histology section); (**C**) The histological score of jejunum tissues. All data are shown as mean ± SEM. Student’s t-test was performed. Statistical significance is indicated by *** *p* < 0.001.

**Table 1 ijms-24-03045-t001:** The statistics of isolates from the chicken intestine.

Medium	Number of Isolates	Number of Uncultured Isolates	Genus	Species
M1	19	5	5	9
M2	110	20	6	17
M3	209	63	4	12
M4	27	0	6	10
M5	17	0	2	3
M6	6	0	2	3
M7	24	0	1	1
M8	15	0	2	5
M9	24	0	1	5
M10	35	0	4	7
M11	21	1	4	7

**Table 2 ijms-24-03045-t002:** Antibiotic sensitivity test of the strain YPG14.

Antibiotics	Disc Conc (μg)	Diameter (mm)	Sensitivity/Resistance
Penicillin	10U	46	S
Ampicillin	10	38	S
Cefalexin	30	43	S
Cefuroxime	30	34	S
Ceftriaxone	30	44	S
Cefoperazone	75	41	S
Chloramphenicol	30	31	S
Gentamicin	10	12	R
Carbenicillin	100	32	S
Vancomycin	30	-	R

## Data Availability

The 16S rRNA sequence of *Limosilactobacillus reuteri* strain YPG14 is available in GenBank with the accession OP953762 (https://www.ncbi.nlm.nih.gov/nuccore/OP953762, accessed on 5 December 2022).

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
