# Peer review of "Culturing the Chicken Intestinal Microbiota and Potential Application as Probiotics Development"

_ijms, 2023, doi:10.3390/ijms24033045_

Round 1
Reviewer 1 Report
The manuscript is focused to the identification and testing of potentially probiotic properties of lactobacilli cultures obtained from the intestines and feces of chickens. The strain Limosilactobacillus reuteri (formerly Lactobacillus reuteri) YPG14 has proven in this regard also in the experimental infection of Salmonella Pullorum chickens. Although the work brings some new knowledge, it contains a number of weaknesses:
Methodological inaccuracies:
1. Authors should explain/describe the chicken euthanasia procedure. This is addressed in the manuscript by reference to the citation of Dumonceaux TJ et al no. 25 (l. 250), in which no such procedure is mentioned.
2. There appears to be a discrepancy between the microbiota isolation methodology used and the results obtained.
· 507 microbiota isolates were obtained, but it is not clear how many chickens were examined, what their ages were, and which sections of the gut were examined.
· What was the preliminary identification procedure? How many morphologically different colonies from one dish were subcultured? It is incomprehensible that, using 11 different media and anaerobic incubation conditions, mainly members of the Lactobacillaceae family were isolated. Then the results presented in Figure 1 can be quite misleading because there are no representatives of anaerobic bacteria known from earlier publications.
· Authors should consider revising the manuscript to focus on the main results describing the characterization of lactobacilli isolates.
Formal comments:
· The authors would like to update the taxonomic status of individual Lactobacillus species (https://lpsn.dsmz.de/species) according to a valid publication: Zheng J, Wittouck S, Salvetti E, Franz CMAP, Harris HMB, Mattarelli P, O'Toole PW, Pot B, Vandamme P, Walter J, et al. A taxonomic note on the genus Lactobacillus: Description of 23 novel genera, amended description of the genus Lactobacillus Beijerinck 1901, and union of Lactobacillaceae and Leuconostocaceae. Int J Syst Evol Microbiol 2020; 70:2782-2858.
· Correct Salmonella pullorum to Salmonella Pullorum (S. Pullorum) throughout the manuscript.
· For culture media, state the manufacturer of the components and the country of origin.
· L.276 correct 16S RNA
L. 296 Staphylococcus aureus USA300 ??? (ATCC)
Author Response
Thank you very much for your careful review and your constructive and very helpful comments and suggestions. The manuscript has been carefully revised according to your comments and suggestions. Please see our point-by-point responses below.
Please see the attachment.

Reviewer 2 Report
Dear authors!
The article submitted for review is devoted to a very interesting topic of сulturing the chicken intestinal microbiota and potential application as probiotics development. In this study considers the possibility of using Lactobacillus strains as a probiotic effective against Salmonella.
The article has all the necessary sections and corresponds to the profile, goals and objectives of the journal.
I would like to note only a small remark on the data analysis in the article.
In the article demonstrated that «Antibiotic sensitivity testing by using the disc diffusion method showed that L. reuteri strain YPG14 was sensitive to the majority of the tested antibiotics except for gentamicin and vancomycin» (Line 226-229). It is known that antibiotics are often used in livestock farms for the prevention and treatment of diseases. In this regard, is it possible to characterize in more detail the prospects for the use of a strain YPG14-based probiotic in the treatment of animals with antibiotics?
Also in the article supplementary file demonstrated that some strains identified as same (For example, 494 YPG-14 Lactobacillus reuteri strain 45b; 495 YPG-16 Lactobacillus reuteri strain 45b; 500 YPG-28 Lactobacillus reuteri strain 45b). How could you describe this coincidence?
I recommend the article for publication after minor revision.
Author Response
Thank you very much for the endorsement of our manuscript and your constructive advice. We have made revisions accordingly, and the point-by-point responses are listed below.
Please see the attachment.

Reviewer 3 Report
Dear Authors . The present paper describe the whole experiment . However there is some elements which should be improved . The section 4.11 dealing with the management of chickens must be clarified. In line 352 you mention " newly hatched " . Please use days of live . How do you control weight of chickens daily ?
Why do use Salmonella Pullorum . when the real problem today is Salmonella enteridtis
Author Response
Thank you very much for your review, and we are also very grateful for your very helpful suggestions. The manuscript has been carefully revised according to your comments and suggestions. Please see our point-by-point responses below.
Please see the attachment.

Round 2
Reviewer 1 Report
Authors must distinguish font style for serotype designation. Salmonella must be written in italics and Pullorum in normal style throughout the manuscript.
Author Response
Thank you very much again for your review, and we are also sincerely grateful for your very professional and helpful suggestions.
According to your suggestion, the styles of "Salmonella" and "Pullorum" have been modified one by one (including Figures 7 and 8), and at the same time, we have also revised the manuscript carefully and proofread it again.